# Transformer-Based Classification of Transposable Element Consensus Sequences with TEclass2

**DOI:** 10.3390/biology15010059

**Published:** 2025-12-29

**Authors:** Lucas Bickmann, Matias Rodriguez, Xiaoyi Jiang, Wojciech Makałowski

**Affiliations:** 1Department of Computer Science, University of Münster, 48149 Münster, Germanyxjiang@uni-muenster.de (X.J.); 2Institute of Bioinformatics, Faculty of Medicine, University of Münster, 48149 Münster, Germany; 3Department of Genetics, Adam Mickiewicz University, 61-712 Poznań, Poland

**Keywords:** transposable elements, TE classification, transformer architecture, deep learning, bioinformatics web tool

## Abstract

Transposable elements (TEs) are mobile genetic elements that are present in great numbers in the majority of eukaryotic genomes. They are major drivers of genome evolution as they can facilitate chromosome rearrangements, provide mechanisms for genomic shuffling, and contribute to genome expansion, thereby altering genome architecture. Additionally, TEs can also modify gene expression by disrupting regulatory sequences, impairing genes, and promoting the emergence of new sequences. Despite their abundance, identifying TEs is challenging and time-consuming due to their extreme diversity in DNA sequence. Many TE families are ancient, and most of their sequences have become inactive due to accumulated mutations and fragmentation. Consequently, different copies of the same TE can differ greatly from each other, which makes identifying decayed copies particularly difficult. In this work, we employ transformer architecture for TE classification. TEclass2 is an integrated classifier that rapidly predicts TE orders and superfamilies using models built on this advanced machine learning approach. The software is available through a web interface, allowing users to classify sequences into sixteen superfamilies according to the Wicker classification system, or alternatively, users can download the source code to train and build custom classification models.

## 1. Introduction

Transposable elements (TEs) are mobile genetic elements which are present in great numbers in the majority of eukaryotic genomes. For example, TEs constitute up to 46% of the human genome [1] and as much as 85% of the genomes of some plants, such as wheat and maize [2,3]. TEs are major drivers of genome evolution as they can facilitate chromosome rearrangements by acting as recombination hotspots, provide mechanisms for genomic shuffling [4], and contribute to genome expansion [5,6], thereby altering genome architecture. Additionally, TEs can modify gene expression by disrupting regulatory sequences, impairing genes, and promoting the emergence of new sequences [7].

Despite their abundance, identifying TEs remains challenging and time-consuming because they are extremely diverse in a DNA sequence, exhibiting a wide range of motifs, lengths, and structures. Many TE families are ancient, and most of their sequences have become inactive due to accumulated mutations and fragmentation. As a result, different copies of the same TE can differ greatly from each other, making the identification of decayed copies particularly difficult [8].

There are two main strategies for identifying TEs. The first involves performing similarity searches against the genome using libraries of known TE sequences. The second, particularly useful for non-model organisms or for discovering novel TEs, is a de novo approach. In this method, interspersed repetitive sequences are identified directly from the genome and used to build consensus models of candidate TEs, which are then compared to known TE libraries for classification and annotation [9].

Multiple approaches can be used for the classification of TEs depending on structural features, transposition mechanisms, enzymatic machinery, or the evolutionary context [10]. The first proposal for classifying TEs [11] divided them into two classes based on their mechanism of transposition. The TEs that use the reverse transcription of an RNA intermediate are called class I elements or retrotransposons, while the TEs that transpose using a DNA intermediate are called class II elements or DNA transposons. Class I TEs were further divided into two categories, LTR-retrotransposons and non-LTR retrotransposons. LTR-retrotransposons are similar to retroviruses, with open reading frames flanked by long terminal repeats (LTRs), whereas non-LTR retrotransposons have no LTRs or very short terminal repeats and a poly-A tail.

This initial classification did not account for non-autonomous elements such as SINEs and has since been revised and updated multiple times. Two of the most widely used classification systems are those from Repbase [12,13] and Wicker et al. [14]. Both systems introduce additional subdivisions to the original two TE classes, considering factors such as sequence similarity, structural features, and the number of open reading frames (ORFs) and encoded enzymes. The Repbase classification divides TEs into the following three main classes: DNA transposons, LTR retrotransposons, and non-LTR retrotransposons, with further subdivisions resulting in sixty-five superfamilies or clades [15]. In contrast, the Wicker system divides class I elements into five orders (LTR, DIRS, PLE, LINEs, and SINEs), and class II elements into the following two subclasses: subclass 1 (cut-and-paste TEs, including TIR and Crypton elements) and subclass 2 (copy-and-paste TEs, including Helitron and Maverick elements) [14].

The need to identify and classify these large genomic components has led to the development of pipelines for semi-automated TE discovery and annotation. Most commonly used tools classify TEs by comparing consensus sequences to TE databases based on sequence similarity. For example, RepeatModeler uses the DFAM database for TE classification, while REPET uses its own RepetDB, which has a classification hierarchy similar to Repbase. Alternative approaches to sequence homology include tools like PASTEC [16], part of the REPET pipeline, which uses hidden Markov model (HMM) profiles to identify TEs.

Another promising strategy is the use of machine learning, which has shown significant potential in bioinformatics [17]. Its application has become increasingly reliable and feasible due to the rapid growth of genomic data and advances in deep learning algorithms that have improved both speed and accuracy [18,19]. Machine learning techniques can automatically identify patterns and extract information from data, making them well-suited for classifying sequences such as DNA.

The first software to use machine learning for classifying TEs was TEclass [20], which used support vector machines (SVMs) to classify unknown TEs based on the frequencies of tetramers and pentamers in repeat elements. TEs were classified into the following main categories: DNA transposons, LTRs, non-LTRs, and further into LINEs or SINEs. The classifiers, built using TE sequences from Repbase, employed a stepwise binary classification as follows: first distinguishing DNA transposons from retrotransposons, then separating LTRs from non-LTRs among retroelements, and finally distinguishing LINEs from SINEs among non-LTR elements [20].

In recent years, deep learning methods based on neural networks have enabled the development of new tools for TE classification. For example, DeepTE [21], TERL [22], and Terrier [23] use convolutional neural network (CNN) architectures to classify TEs into their respective orders or superfamilies. Some tools are specialized for specific TE orders as follows: TIR-Learner [24] classifies TIR elements using neural networks, k-nearest neighbor, random forest, and Adaboost; TE-Learner [25] uses a random forest approach for LTR elements; and Inpactor2 [26] applies CNNs to classify LTR retrotransposons in plant genomes. In TEclass2, we employ the transformer deep learning model [27], t originally developed for natural language processing (NLP), and computer vision. Transformers have demonstrated exceptional performance in these fields, often surpassing models such as recurrent neural networks (RNNs) [28]. They process entire inputs simultaneously using an attention mechanism that provides context for any position in a sequence, enabling parallelization and reducing training time [27].

Other studies have also applied transformers in bioinformatics, such as DNABERT [29], which is based on the BERT transformer language model [30] and used for predicting promoters, splice sites, and transcription factor binding sites. Similar transformer-based approaches have been used for bacterial promoter classification, viral genome identification, and analysis of mRNA degradation properties in COVID-19 vaccine candidates [31]. Recently, InstaDeep group developed series of tranformer-based models for human genomics [32].

While vanilla transformer and BERT models achieve state-of-the-art performance in their respective fields, they have a runtime and space complexity of O(n^2^), typically limiting them to processing sequences of up to 512 tokens. To address this, linear transformer models such as Linformer [33] and Longformer [34] have been developed, allowing processing of sequences up to 4096 tokens or more, depending on available memory. However, these architectures are not yet fully adapted for continuous, unseparated sequences like those found in DNA.

In this work, we introduce a new architecture based on the Longformer model [34] for classifying selected TE sequences, incorporating sequence-specific augmentations, a k-mer specialized tokenizer, and sliding window dilation. TEclass2 is an integrated classifier that rapidly predicts TE orders and superfamilies using models built on this advanced transformer architecture. The software is available through a web interface, allowing users to classify sequences into sixteen superfamilies according to the Wicker classification system [14], or alternatively, users can download the source code to train and build custom classification models.

## 2. Methods

### 2.1. Transformer

A transformer is a deep learning model that uses the mechanism of self-attention originally designed to handle sequential input, especially natural language, and is vastly improved in its performance when compared to its predecessors, the recurrent neural networks (RNNs) [35] and long short-term memory (LSTM) [36]. Instead of using the input as a sequence and processing each word in order as RNNs do, they process the whole sentence concurrently, allowing parallel computation and improving long-term dependencies. Originally transformers were designed using sequence transformation with a Seq2Seq model for performing tasks like translation and summarization of text. This model has an encoder that captures the input and passes it to a decoder that produces an output [37].

In TEclass2, for the classification of TE DNA sequences, we use only the encoder-block, followed by a classification head as in a linear layer (Figure 1). The transformer model is divided into several layers which have multiple attention heads [38] and each one can learn independently relevant parts of the input sequence.

The first layer’s query is the encoded query or tokens, and the following layers are based upon previously computed attention matrices as queries. In order to take into consideration the order of the input sequence, we added positional encoding. Due to memory constraints, transformers can process up to 512 tokens, and larger inputs are either truncated or processed in chunks. For the latter the classification is then computed as the median of the output from each of the processed chunks. To solve this problem, different methods have been proposed, such as the Longformer method, which tackles this with reduced self-attention masks of local attentions and by adding some global attentions [34].

A way to increase variants in the data during training is to use data augmentation and force the machine learning model to adapt to more generalized features [39]. To apply an on-the-fly sequence augmentation during the training step, we developed a variety of biologically motivated data augmentations (Table 1).

Depending on the applied augmentations, the DNA sequence may also differ in length. The only transformation that introduces ambiguity is the introduction of Ns in the sequence; in this case, the transformer should learn that these characters do not represent relevant information. The original transformer does not handle information about the position of tokens due to the transformers’ parallel processing of the data; therefore, the relationship between tokens has to be provided, as these distances provide important spatial information. Sinusoidal positional encoding is used for this as it has the advantage of providing information that is independent from the length of the input and each value ranges between zero and one [40].

In TEclass2 the DNA sequences used as an input undergo a process known as tokenization, in which the input strings are split into smaller parts called tokens, which are assigned identifiers based on created vocabulary. Tokenizers can also be embedded into a vector space and used as a form of compression and dictionary coding. We used k-mer tokenization with a sliding window approach dividing each DNA sequence into k-mers, whereas the general length of the embedded string does not decrease when using a different window size. A larger window size dilates the steps between the k-mers, which increases the coverage of the input sequence but reduces the overlap and dependencies between them, as they also have the advantage of generating a fixed number of words. In our software we used k-mers with a length of five with a sliding step of two nucleotides. The number of DNA 5-mers including unknown nucleotides as N gives a rather small vocabulary of 3125 tokens. With this software it is possible to use k-mers of different length but it is important to consider that using the shorter ones impedes the encoding of different semantic representations, and using the larger ones increases the size of the vocabulary manifold degrading the performance [41].

In order to allow long DNA sequences, we introduced a dynamically scaling sliding step of minimal two nucleotides to maximize the coverage of the input sequence, while trying to retain overlapping tokens. We compute it as:$$step\_size=min\left(\left[\frac{len\left(seq\right)}{max\_embedding\_size}\right],k-1\right)$$

The attention values are traced back to each token and using linear interpolation these values are scaled back. This reduces cross-dependencies between local tokens and increases long-range dependencies, which are useful to classify long TE sequences.

The models tested were trained with weighted cross entropy (WCE), as the datasets are unbalanced. The Softmax function [42] is used to compute the probabilities for predicting the TE classes; to classify the performance of each trained model we used different metrics during evaluation and testing. This includes the train- and evaluation-loss, class-wise precision, recall, F1 score, and global accuracy on the test dataset; also, a macro average and weighted average were computed for precision, recall, and F1 score.

### 2.2. Workflow

The input data for training and building a model with TEclass2 are a FASTA file with the TE models, where the header of all sequences contains the name and the class of TE. For the training of a model, the workflow (Figure 1) starts by building a database of labeled TE sequences, which is divided into training, evaluation, and testing datasets. The configuration of the database is read from the configuration file, where the expected family names and file paths must be specified.

After the database has been built, the training step, used to train the model, can begin utilizing parameters from the configuration file, such as the number of epochs, the size of the layers for the neural network, size of the embeddings, window dilation, and many others. The training step uses the data augmentation properties specified in Table 1, which increase the variability of the input sequences; then, the TE sequences are scanned using a sliding window approach.

Next, each k-mer is tokenized and used as an input for the transformer algorithm that computes the local attention for the whole sequence and the global attention for certain specific positions. After this step, the TE sequences that were not used for training are evaluated with the transformer model, and a TE class is predicted and compared with the labeled data. During the training, a checkpoint is saved after each epoch, which allows the user to restore the training process from a previous step in case the training needs to be restarted or extended. In the final step, the trained model is validated on the test-set, which consists of unseen and unbiased comparable data.

For the classification of TEs, we use sliding windows to compute the k-mers of the input sequence and tokenize these data for use with the transformer model. The model then outputs a prediction score, which is normalized with the Softmax function, and gives the probability of the input sequence of belonging to any of the classes specified in the model.

### 2.3. Datasets

In the previous version of TEclass, the software permitted the classification of TEs into four categories, namely SINEs, LINEs, LTRs, and DNA transposons. To enhance the classification capabilities and allow a more in-depth categorization of TEs and by taking advantage of the larger datasets available today, we built a classification model that assigns TEs to sixteen different categories of TEs. These categories are based on the Wicker classification and include eight orders from class I (RNA-mediated transposons), i.e., Copia, Gypsy, Pao, ERV, L1/L2, RTE, jockey, and SINEs, and eight orders from class II (“cut-and paste” transposons), i.e., Tc1/Mariner, hAT, Merlin, Transib, P, Crypton, Helitrons, and Mavericks (Polintrons) (see Table 2).

For the creation of the curated dataset for training the models, we used TE sequences from Dfam, curated and non-curated version 3.7, and the curated Repbase version 18. In addition, the curated database was very limited and too small for models that are more complex. To allow more categories of TEs, we need a larger dataset with a larger variety of TE classes. The use of a non-curated database improves the number of data points, albeit with a slight decrease in the quality of the results. From all the TE sequences in the dataset, 75% are used for training the model, 15% are used for validation, and 10% for testing (Table 2).

### 2.4. Evaluation of TE Models

In order to evaluate the performance of the trained models, we calculated the precision, recall, and F1 score of each one of the TE classes of a model to assess its reliability in the classification. Precision is a measurement of the exactness of the model and is the number of true positives divided by the sum of true positives and false positives. A high value of precision indicates that false positives are rare in the model. Recall is a measurement of the ability of the model to identify all the correct predictions and is defined as the proportion of true positives divided by the sum of true positives and false negatives. A high value of recall indicates a good level of completeness of the classification model, meaning that false negatives are not common. Although both measures are useful, they have their own limitations, as precision does not consider misclassifications while recall does not consider false positives. The F1 score combines both measures as it is the harmonic mean of precision and recall, where both contribute equally to the score, which has values between 0 and 1.

Since in each model we are using multiple classes with different numbers of TEs for comparing whole models between themselves, we calculated the F1 weighted average using the following formula:$${F1}_{w-avg}=\sum \left(2\left(\frac{precision*recall}{precision+recall}\right)*\frac{n\_obs\_TE\_class}{n\_total\_TEs}\right)$$

The weighted F1 score takes into consideration the class imbalance of the models and calculates the F1 score for each TE class, which is then multiplied by the proportion of TEs from that class in the total dataset. During the training of the models, these scores are calculated after each training cycle or epoch, as we used 10% from each TE class of the dataset for testing the performance of each model, as shown in the column “Test” from Table 2. In this way, during the training the trajectory of the changes in the performance can be checked in real-time. We also evaluated the performance of the software by classifying TEs models obtained from annotated genomes.

### 2.5. Software and Hardware Specifications

The hardware used for training were an Nvidia GTX 1660Ti 6 GB, RTX 3080 12 GB, A100 40 GB, and A100 80 GB. Palma II (https://palma.uni-muenster.de/documentation/; accessed on 15 October 2025), the HPC of the University of Münster (Münster, Germany), provided the latter two. These tests were conducted using float32 training, as mixed precision training is still marked as an experimental feature in the Trainer API. Gradient accumulation was used for models based on non-curated data or very complex parameters, to overcome memory constraints. TEclass2 was developed using Python 3.10 and requires NumPy (we used version 1.21) [43], scikit-learn (we used version 1.0) [44], PyTorch (we used version 1.12) [45], Tensorboard (we used version 2.4) [46], transformers, and tokenizer packages [28].

## 3. Results

### 3.1. Parameters Used for Training TE Models

During the training of the models, we evaluated different configurations, pre-processing steps, and parameters regarding the complexity of the models. We tested building models using 4-mer and 6-mer, but this resulted in a very similar performance to the default 5-mer models. As expected, allowing longer sequence inputs by increasing the embedding size provides an overall improvement of the results, in particular for longer TEs. We noticed the importance of using dilated sliding windows to improve the results when the input contained long sequences. This resulted in an increased F1 score, which impacted the overall performance of the trained model.

Increasing the sparse attention did not improve the precision, indicating that local attention did not have an impact on classification beyond a specific minimal attention window and increasing it did not produce significant improvements. We also noticed that the complexity of each layer beyond a certain point did not increase the performance metrics, but what affected the most was the complexity of the model itself. In contrary to other methods of model complexity, the use of additional global attention slightly increased the overall scores.

To summarize, the complexity of the model and the available data points did not change the performance as much as expected. This introduced grain fine-tuning with different parameters for TEclass2 (Table 3). The classifier approach using 5-mers with dilated sliding windows, using the maximum embedding size with 2048 tokens alongside a few additional global attention tokens, achieved the highest scores.

### 3.2. Training of TE Models

We built a transformer model that allows the classification of TEs in sixteen categories; the performance results with the training dataset can be observed from a confusion matrix (Figure 2) and a table of model testing statistics (Table 4). For classifying TEs beyond just their main order, as the previous version of TEclass does, the data from curated databases were insufficient, so we relied on the Dfam version 3.7 non-curated dataset and Repbase version 18. We acknowledge the risks involved in using non-curated data, yet we expect that the majority of the sequences will be correctly classified, outweighing biases introduced by those erroneously labeled.

As expected, when more categories are added in order to train a model, the model increases in complexity, takes more time to train, and there is a greater chance of misidentification. Furthermore, some classes of TEs are similar to each other, making classification difficult. When we train a model using sixteen different classes of TEs, we observe good performance, considering the size of the model for the hAT, L1/L2, and ERV transposons, all obtaining F1 scores greater than 0.8. The performance of the model in classifying the TcMar, SINEs, RTE, Transib, Helitrons, and Gypsy transposons shows F1 scores for all of them greater than 0.7. The performance with some other transposons, like Copia, Crypton, Maverick, Merlin, and P, was suboptimal, with F1 scores below 0.7. In general, the weighted average of the model has an F1 score of 0.76, with a precision score of 0.77 and a recall of 0.76. The misclassification that we observed can be attributed in part to the limitation of the transformer model itself or to how the models were built, but it can also be attributed to mislabeled data in the TEs dataset or to the lack of enough representative sequences for training the model in a given class of TEs.

### 3.3. Comparison with Other Machine Learning Tools

To assess the classification performance of the TE model trained with TEclass2, we employed TE models built using RepeatModeler2 [47] version 2.0.3 with default parameters and the LTR-detector module on the fruit fly and rice genomes, both obtained from the UCSC Genome Browser database. These genomes were chosen because they have high-quality, mostly complete TE annotations, which makes the benchmark more reliable. The two species also represent very different taxa, allowing evaluation of the model across diverse types of TEs. RepeatModeler already provides TE model classifications through its RepeatClassifier module. To further curate these results, we ran nhmmscan from the HMMER package version 3.3.2 (hmmer.org), with the parameters “--noali -E 1e-10” to align these TE models with the curated Dfam database version 18. Sequences that exhibited poor alignment with TEs or had hits to TEs from different orders were discarded, as this often indicates chimeric models, a common problem in most TE model building software [8].

We ran the TE models obtained from the rice and fruit fly genome with TEclass2 and compared them to the classification assigned by RepeatModeler. We also ran these TE models using the default settings of two other machine learning softwares, DeepTE and TERL. We performed benchmark comparisons with these two tools because they both use deep learning strategies to classify TEs at the superfamily level and cover multiple TE superfamilies that overlap with those in our dataset. This makes them the most comparable tools to TEclass2. Making a direct comparison of TE classification tools can be difficult, as some of them specialize in solving specific problems, such as differentiating TEs within the same class or classifying into a different number of families. For the 75 TE models built from the rice genome, most of them are Gypsy and Copia, but there are also some Helitrons and LINEs. TEclass2 using a minimum probability threshold of 0.7 was able to classify 79% of the TEs models; from those, 82% were identified correctly at superfamily level and for 87% the order was also predicted accurately.

For the fruit fly genome, 667 TE models were built, with a diverse number of TE models from Gypsy, Jockey, Bel-Pao, R1, Tc1-Mariner, Copia, P, Helitron, Transib, and hAT. From this dataset, 393 were classified with a probability of at least 0.7, that is 59% of the total, and from those 285 were correctly classified, meaning a 73% accuracy (see the Data Availability Statement). Adopting a stricter probability threshold, such as 0.9, improved the TE prediction accuracy to 83%, but reduced the number of TEs that are classified to 281 out of 667. During these analyses there was also a small group of TEs that could not be classified because they do not belong to any of the families included in the trained model.

For testing other machine learning tools, we ran DeepTE with the parameters “-m M -sp M” for the fruit fly TEs models in order to specify that TEs are from a Metazoa genome and “-m P -sp P” for rice TEs models to specify that the TEs are coming from a plant genome. By default, DeepTE sets a limit of a probability of 0.6 to give a result; if the value is lower, then the TE is not classified. For the TEs models built from the rice genome, the accuracy for predicting the TE order was 53%, and 45% for the superfamily. From all the TEs models built, only 79% had their order predicted and 29% their superfamily. For the TE models built from the fruit fly genome, DeepTE was able to predict the order accurately for 35% and the superfamily for 32%. The proportion of TEs with the order predicted was 88% and for the superfamily 58% (see the Data Availability Statement).

We also ran TERL for both genomes, using the model DS3, which is a model that can classify TEs into twelve superfamilies. In the case of TE models constructed from the rice genome, the accuracy of predicting the TE order was 68%, and 50% for the superfamily. Among all of the TE models constructed, a classification rate of 91% was achieved, as 9% were classified as non-TEs. For TE models from the fruit fly genome, 42% were accurately predicted and 21% for the superfamily. Approximately 57% of the TEs had their order predicted (see the Data Availability Statement at https://github.com/IOB-Muenster/TEclass2/tree/main/tests). A summary of the benchmarking experiments is provided in Table 5.

### 3.4. TEclass2 Web Interface

The models built with curated and non-curated data can be used to classify TEs at https://bioinformatics.uni-muenster.de/tools/teclass2/index.pl.

The web interface provides a simple interface that allows the classification of TEs models DNA sequences into sixteen superfamilies and the possibility to filter the results by a probability value. The TE sequences used as input can be uploaded as a FASTA file or input in the text box.

The output produced by the classification module provides an accessible and compact format as a table or as a downloadable TSV file useful for further analysis. This table includes the classification predicted, with the order and superfamily that corresponds to the class with the highest probability, and the Softmax confidence scores for each class in the model (Figure 3). For training models, it is necessary to download the source code of the program from https://github.com/IOB-Muenster/TEClass2 and follow the configuration instructions in order to run it. The trained models for TEclass2 can be downloaded directly from https://bioinformatics.uni-muenster.de/tools/teclass2/models/teclass2-models.tgz.

### 3.5. Known Limitations

Some of the limitations of TEclass2 are the hardware requirements for training new models, as it is recommended to use a GPU to build new models. However, principal models can also be built using CPU-based machines. Other weaknesses come directly from the transformers’ architecture [48], as the training of models require a fine-tuning of multiple parameters, which may lead to several trial and error cycles. Inherent to the architecture is the difficulty to control the attention mechanism, which can put attention in parts of the sequences that are biologically irrelevant or show an undesirable attention-bias. The algorithm also needs to fragment the sequences into k-mers, meaning that the sequence is not analyzed as a whole, but divided into words, similarly to NLP inputs.

Another issue may come from the input data itself, as machine learning strategies require a large number of TE sequences that are well curated. This may be impossible to attain for some classes of TEs, where only a few copies are present in curated databases. It should also be emphasized that Dfam is highly biased toward model organisms. It was built around the following five species: human, mouse, zebrafish, flatworm, and fruit fly. This bias persists to these days with most of TE models in curated part of Dfam coming from mammalian genomes. Uncurated part is slightly better, although it comes with the price that some of the models may be misclassified. Dfam 3.7 included 15,415 curated TE families’ models and 1,037,077 uncurated families from 2346 taxa. For comparison, the current version of Dfam (3.9) consists of over four million models from 2784 taxa (https://www.dfam.org/releases/Dfam_3.9/relnotes.txt). To improve TEclass2 performance, more diverse data should be included during the training step. Such data may come from the newer Dfam release and/or other TE databases such as FishTEdb (https://www.fishtedb.com).

Several tools for TE classification that use a machine learning approach have recently been developed. Interestingly, TEClass2 is the only all-in-one, flat classifier, and its classification is based purely on the transformer architecture. All other methods rely on CNNs and RNNs. TEClass2 is also the only architecture that captures the entire sequence without truncating or fragmenting it into multiple parts. We summarized the main feature of these tools in Table 6.

## 4. Conclusions

TEclass2 is a comprehensive classifier that uses state-of-the-art transformer architectures to produce promising results. It achieves an overall weighted accuracy of 79% across sixteen TE families. In addition to its performance as a standalone tool, TEclass2 is a fully open source and designed to be a flexible framework for the research community. We provide the complete codebase to enable immediate classification via our pre-trained models and to allow researchers to train their own custom models or fine-tune existing models on specific taxonomic datasets. We hope this accessibility will empower users to explore the strengths and limitations of transformers in genomics and ultimately improve TE classification methods, regardless of performance outcomes on novel datasets. It would also be interesting to see how the models improve over time with better training datasets. We acknowledge that the training data are imperfect, as both Repbase and Dfam are biased toward model organisms. Consequently, we are eager to see how our system will perform with improved training datasets.

## Figures and Tables

**Figure 1 biology-15-00059-f001:**
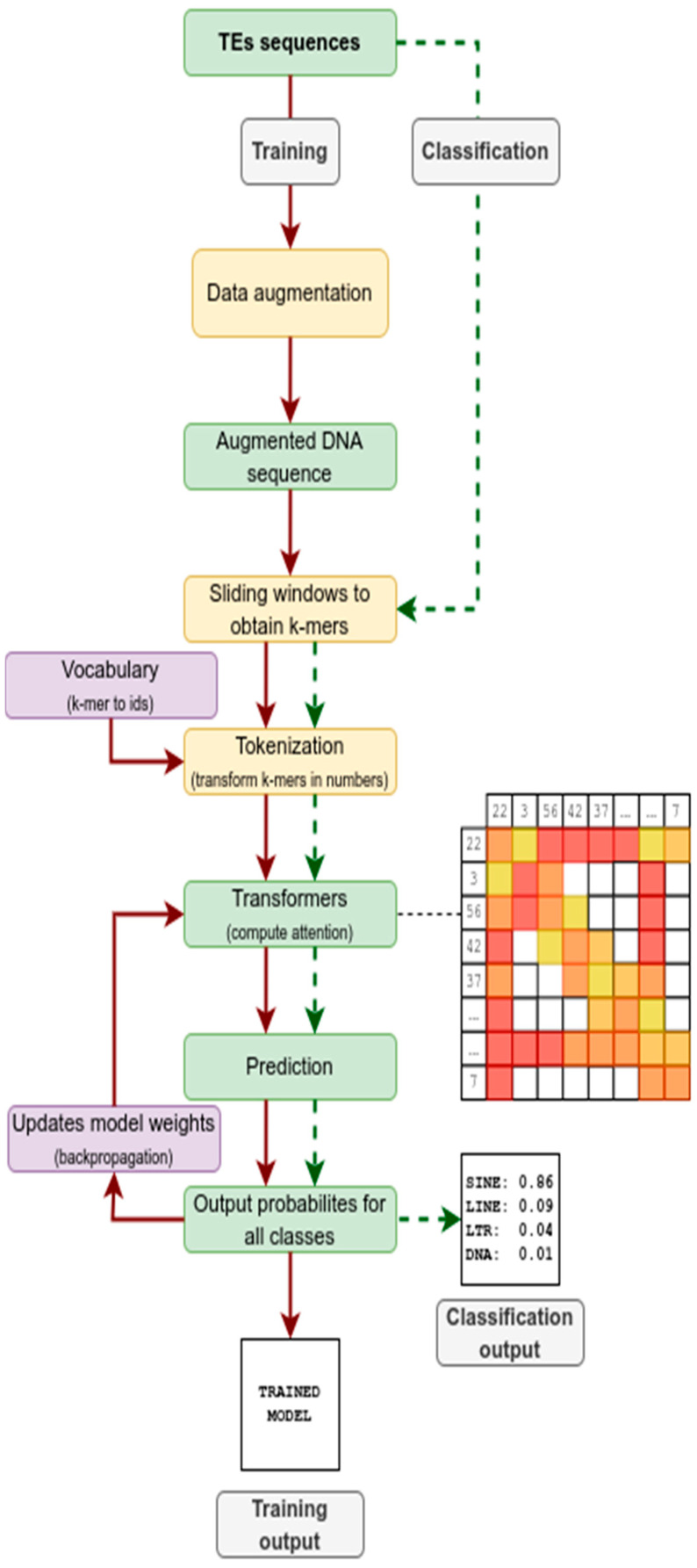
Flowchart succinctly describing how TEclass2 works both in the training of dataset to produce transformer models (red arrows) and the use of these models to classify TE sequences (dotted green arrows).

**Figure 2 biology-15-00059-f002:**
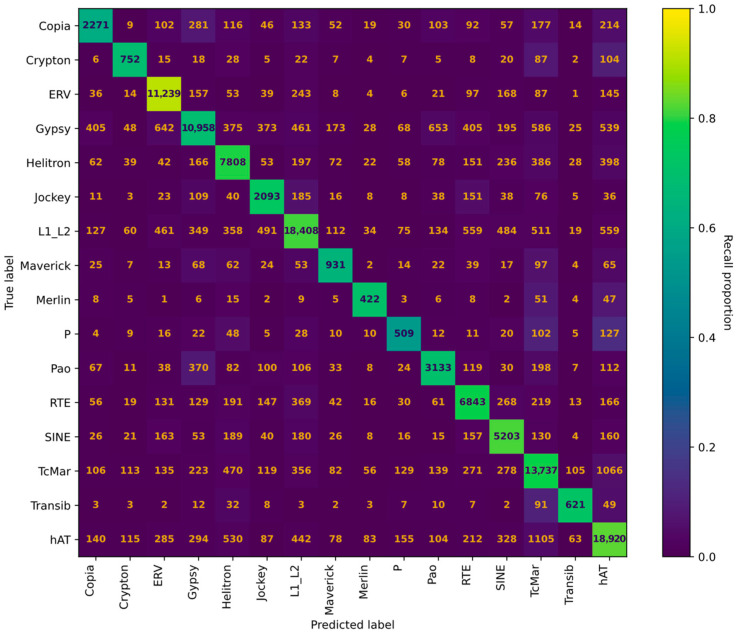
Confusion matrix for the training of non-curated data from Dfam version 3.7 and Repbase version 18. Each row represents a different TE type, and each column includes number of models as predicted for each category. For example, of the 3716 Copia element models analyzed, 2271 were classified as Copia, 9 as Crypton, 102 as ERV, and so forth. Additionally, cell colors represent a heat map based on recall values for each TE category. Darker colors represent lower recall values and consequently poorer performance of the classifier in the given TE class.

**Figure 3 biology-15-00059-f003:**
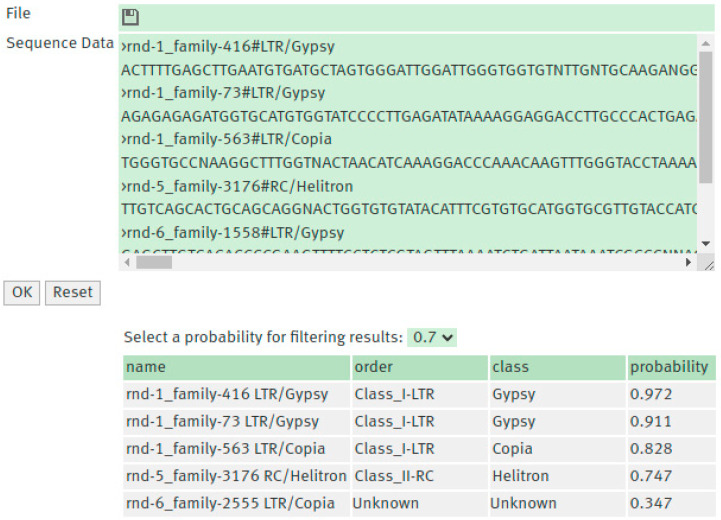
The web interface for TEclass2 allows inputting the TE sequence to be classified in a text box or as a FASTA file. The output shows the classification and the Softmax values the model scored for each class of TE.

**Table 1 biology-15-00059-t001:** List of data augmentation used in DNA sequences for the data-training step in TEclass2.

Augmentation	Description
SNP	Replace randomly a single nucleotide.
Masking	Replace a base with an N.
Insertion	Insert random nucleotides in a sequence.
Deletion	Delete a random part of a sequence.
Repeat	Repeats a random part of the sequence.
Reverse	Reverses the sequence.
Complement	Computes the complement sequence.
Reverse complement	Computes the opposite strand.
Add tail	Adds poly-A tail to the sequence.
Remove tail	Removes poly-A tail if present.

**Table 2 biology-15-00059-t002:** DFAM version 3.7 curated and non-curated data and Repbase version 18 used for training TEclass2, showing how the TEs from different categories were used for training, validation, and testing.

Group	Training	Validation	Test	Total
Copia	18,584	3717	2478	24,779
Crypton	5453	1091	727	7270
ERV	61,539	12,319	8212	82,124
Gypsy	79,674	15,935	10,623	106,232
hAT	114,707	22,941	15,294	152,943
Helitron	48,980	9796	6531	65,307
Jockey	14,201	2840	1894	18,935
L1/L2	113,708	22,742	15,161	151,610
Maverick	7218	1444	962	9624
Merlin	2971	594	396	3961
P	4692	938	626	6256
Pao	22,192	4438	2959	29,589
RTE	43,500	8700	5800	58,000
SINE	31,960	6392	4261	42,613
TcMar	86,925	17,385	11,590	115,900
Transib	4279	856	571	5705
**Total**	**660,636**	**132,127**	**88,085**	**880,848**

**Table 3 biology-15-00059-t003:** Optimal parameters used to train the transformer model with both curated and non-curated dataset. These parameters may not be optimal for other datasets and their values depend on the hardware resources available.

Parameters	Values	Description
num_epochs	100	Times the model is trained on the same data.
max_embeddings	2048	Maximum length of the input sequence but can be overcome with the Longformer model.
num_hidden_layers	8	Dimensions of the encoder hidden state for processing the input.
num_attention_heads	8	Number of heads (neurons) used to process input data and make predictions.
global_att_tokens	[0, 256, 512]	Positions of the input where the entire sequence is considered.
intermediate_size	3078	Dimensionality of the feed-forward layer.
kmer_size	5	Length of the words for dividing the input sequence.

**Table 4 biology-15-00059-t004:** Evaluation parameters obtained from curated dataset using sixteen TE classes.

Group	Precision	Recall	F1-Score	Support
Copia	0.69	0.60	0.64	3716
Crypton	0.62	0.69	0.65	1090
ERV	0.86	0.91	0.88	12,318
Gypsy	0.81	0.71	0.76	15,934
hAT	0.84	0.82	0.83	22,941
Helitron	0.72	0.81	0.76	9796
Jockey	0.61	0.71	0.66	2840
L1/L2	0.89	0.80	0.84	22,741
Maverick	0.56	0.64	0.60	1443
Merlin	0.64	0.70	0.67	594
P	0.35	0.60	0.44	938
Pao	0.71	0.70	0.70	4438
RTE	0.76	0.78	0.77	8700
SINE	0.76	0.80	0.78	6391
TcMar	0.76	0.77	0.76	17,385
Transib	0.67	0.75	0.71	855
Accuracy			0.79	132,120
Macro avg	0.70	0.74	0.72	132,120
Weighted avg	0.79	0.79	0.79	132,120

**Table 5 biology-15-00059-t005:** Comparison of deep learning tools for TEs classification on rice and fruit fly models, showing prediction accuracy at the order and superfamily levels.

Genome	Number of TE Models	Tool	% Classified	Accuracy (Order)	Accuracy (Superfamily)
**Rice**	75	TEclass2	79%	87%	82%
		DeepTE	79%	53%	45%
		TERL (DS3)	91%	68%	50%
**Fruit fly**	667	TEclass2	59%	79%	73%
		DeepTE	88%	35%	32%
		TERL (DS3)	57%	42%	21%

**Table 6 biology-15-00059-t006:** General comparison of TE classifiers that use deep learning approach.

Comparison Aspect	TEclass2	DeepTE	CREATE [49]	BERTE [50]	Terrier
**Classification Architecture**	Transformer	Convolutional neural network	Convolutional neural network + recurrent neural network	Convolutional neural network	Convolutional neural network
**Features/Tokenization**	Sliding window k-mers (whole sequence)	K-mer counts	Global k-mer frequencies and terminal motifs (LE/RE)	Cumulative k-mer frequencies + BERT Encoder embedding of truncated sequences	Raw nucleotide sequences with a CNN kernel size of 7
**Classification Structure**	Flat	Hierarchical	Hierarchical	Hierarchical	Hierarchical
**Strengths**	Captures long-range dependencies; takes the whole sequence into account	Fast inference	Balances global patterns and terminal motifs		Uses the 29.10 release of Repbase with over 100,000 repeat families for training

## Data Availability

Precomputed TE models are available at https://bioinformatics.uni-muenster.de/tools/teclass2/models/teclass2-models.tgz. Curated models of TEs for the fruit fly and rice genomes are available at https://github.com/IOB-Muenster/TEclass2/tree/main/tests.

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
