# Peer review of "Transformer-Based Classification of Transposable Element Consensus Sequences with TEclass2"

_biology, 2025, doi:10.3390/biology15010059_

Round 1

Reviewer 1 Report

Comments and Suggestions for Authors

In this manuscript, the authors present TEclass2, a deep-learning–based tool using a linear Transformer architecture for the classification of transposable element (TE) consensus sequences into 16 superfamilies defined according to the Wicker et al. classification system. The topic is relevant to the TE community and more generally to the genome annotation field since the classification of TE consensus is a really important step during TE annotation in complete genome assemblies. The manuscript describes a potentially valuable tool which is an enhancement of the previous version published in 2009. The manuscript is generally clearly written, but several issues need to be addressed before publication.

Major comments

1) The manuscript relies heavily on non-curated Dfam data to generate the training dataset, which may introduce systematic annotation noise. Although the authors acknowledge this, the consequences are not sufficiently examined. Maybe it would be informative to provide quantitative estimates of mislabeling rates or dataset inconsistencies, to evaluate model robustness to labeling noise using subset analyses (e.g., training on curated-only vs mixed datasets) and discuss whether certain low-performance classes (e.g., P, Maverick, Copia) correlate with poorly curated categories.

2) Related to the previous point, it is known that both Dfam and Repbase are biased in the representation of some species. Thus there could be some species-specific biases that could noise the training. Moreover any imbalance in the representation of particular TE types could also contribute to the poor performance observed for some TE classes. This should be examined thoroughly.

3) Regarding the benchmarking section, line 351 (p. 11) mentions the use of the fruit fly and zebrafish genomes, yet the subsequent sentences refer to rice and fruit fly. This inconsistency is confusing and should be clarified.

4) The choice of the two organisms used for the benchmark should be justified as well as the comparison to only TERL and DeepTE.

5) All the results presented in the section 3.3 would be clearer with a summarized presentation as a table or a figure.

Minor Points

1) Figure 2 needs a more descriptive legend (especially to explain the scale on the right)

2) The results section would benefit from better organization, especially separating performance optimization, evaluation, and benchmarking.

Author Response

Detail responses to the reviewers

Reviewer 1

In this manuscript, the authors present TEclass2, a deep-learning–based tool using a linear Transformer architecture for the classification of transposable element (TE) consensus sequences into 16 superfamilies defined according to the Wicker et al. classification system. The topic is relevant to the TE community and more generally to the genome annotation field since the classification of TE consensus is a really important step during TE annotation in complete genome assemblies. The manuscript describes a potentially valuable tool which is an enhancement of the previous version published in 2009. The manuscript is generally clearly written, but several issues need to be addressed before publication.

Thank you for your input, it is greatly appreciated.

Major comments

1) The manuscript relies heavily on non-curated Dfam data to generate the training dataset, which may introduce systematic annotation noise. Although the authors acknowledge this, the consequences are not sufficiently examined. Maybe it would be informative to provide quantitative estimates of mislabeling rates or dataset inconsistencies, to evaluate model robustness to labeling noise using subset analyses (e.g., training on curated-only vs mixed datasets) and discuss whether certain low-performance classes (e.g., P, Maverick, Copia) correlate with poorly curated categories.

That’s an excellent idea, which we would love to explore. Unfortunately, due to the reasons mentioned above, it will have to wait until the next major update of the software.

2) Related to the previous point, it is known that both Dfam and Repbase are biased in the representation of some species. Thus there could be some species-specific biases that could noise the training. Moreover any imbalance in the representation of particular TE types could also contribute to the poor performance observed for some TE classes. This should be examined thoroughly.

That’s a valid point indeed. We will try to explore the issue further during the next models’ update. As soon as we have the necessary personnel, we will include data from other more specialized databases, such as FishTEDB, plant-focused databases, etc. 

3) Regarding the benchmarking section, line 351 (p. 11) mentions the use of the fruit fly and zebrafish genomes, yet the subsequent sentences refer to rice and fruit fly. This inconsistency is confusing and should be clarified.

“Zebrafish” was a typo and it has been corrected to “rice,” thank you very much for pointing this out!

4) The choice of the two organisms used for the benchmark should be justified as well as the comparison to only TERL and DeepTE.

The main reason why we chose these genomes (fruit-fly and rice) is that they have high quality TE annotations, which ensures that our benchmark is more reliable. Both species have mostly complete and well-curated TE annotations, and belong to very different taxa, allowing us to evaluate the model across a diverse type of TEs.

Regarding the comparison with TERL and DeepTE, these are the deep-learning TE classifiers most directly comparable to our approach. Both tools classify at the superfamily level and cover multiple TE superfamilies making them suitable for this benchmark.

5) All the results presented in the section 3.3 would be clearer with a summarized presentation as a table or a figure.

We added a table to summarize this section. We hope it is more clear now.

Minor Points

1) Figure 2 needs a more descriptive legend (especially to explain the scale on the right)

This figure was modified for better clarity and the legend was improved at the same time.

2) The results section would benefit from better organization, especially separating performance optimization, evaluation, and benchmarking.

We tried different variants of the Results presentation but in our opinion it didn’t improve the overall flow of the presentation. Consequently, we stayed with the original presentation style. However, we have rewritten the text in a few places, which hopefully improved clarity of our discourse. 

Reviewer 2 Report

Comments and Suggestions for Authors

I have now reviewed the manuscript of Bickmann and collaborators entitled "Transformer-based classification of transposable element consensus sequences with TEclass2". In this work the authors present a machine learning based classifier for transposable elements consensus sequences. 

This is a very important and challenging topic in the field of TEs that requires attention, and deep learning is a promising avenue that need to be explored in depth. The current limitations to applying AI methods to TE classification need to be clearly highlighted in order to educate users to the potential risks of using immature methods and avoid misinterpretation of biological data.

Unfortunately, after carefully reviewing the training data I believe that in its current form the model present a risk to disseminate errors while performing automatic classification if this method is used without expert knowledge of the data used for training.

Indeed, a large chunk of the training data include the non-curated section of DFAM 3.7 which are not suited in my opinion to train a classifier. This section of DFAM is essentially composed of raw, unreviewed RepeatModeler runs. The consensus in these library can be complete (full-length element with exact boundaries), but in practice these libraries have redundancy, are fragmented, contain chimerical elements and contamination from gene families and non TE repetitive elements. All of these are classified with Repeatclassifier, an homology-based module that relies on the local DFAM installation present on the machine where the run is executed. The version and scope of the local DFAM library can vary widely (DFAM libraries are divided in partitions that can be individually installed). This means that the quality of the label attributed to the non-curated DFAM consensus is expected to be highly variable and in my opinion no suited to train a TE classifier.

Nevertheless, this should not prevent the authors to explore the current limitation of using a transformer based framework for TE classification. Whatever the results are without the un-curated DFAM consensus, the science behind it will be valid and will inform further work in the field. We can perhaps learn what classes or types of TEs are more likely to be recognized by such model architecture and identify areas were progress need to be made. Note that DFAM is now at 3.9 and more curated models have been added.

There are also interesting points in this manuscript that need to be highlighted. I think that the evaluation experiment has some merit by recreating realistic use cases of building de novo libraries with Repeatmodeler and attempting classification. However, the procedure should be described more precisely: how does the HMMER parameters selected by the authors translate in terms of confidence in the classification? For instance, I would feel more confident with controls on how much a consensus is covered by a hit with a certain percent of identity to a curated DFAM element. Perhaps retaining best reciprocal matches between the Repeatmodeler2 library and DFAM would create a high quality set to evaluate?

Recently, a CNN-RNN method was published but failed to compare their results to TEclass2 (even though it was mentioned in introduction); including CREATE (https://academic.oup.com/bib/article/26/6/bbaf608/8324532) to the evaluation would be very informative. Overall it would be great to have a figure summarizing the comparisons between ML methods.

Finally, I think that the most important outcome of this work would be advising potential users on how to make the best use of their model in further research, whether in ways to improve the model or ways to use the model to improve classification of TE. I would like to stress that in my view it doesn't matter how good the model perform. Even negative results are informative for the TE community.

Author Response

Reviewer 2

I have now reviewed the manuscript of Bickmann and collaborators entitled "Transformer-based classification of transposable element consensus sequences with TEclass2". In this work the authors present a machine learning based classifier for transposable elements consensus sequences.

This is a very important and challenging topic in the field of TEs that requires attention, and deep learning is a promising avenue that need to be explored in depth. The current limitations to applying AI methods to TE classification need to be clearly highlighted in order to educate users to the potential risks of using immature methods and avoid misinterpretation of biological data.

Unfortunately, after carefully reviewing the training data I believe that in its current form the model present a risk to disseminate errors while performing automatic classification if this method is used without expert knowledge of the data used for training.

Indeed, a large chunk of the training data include the non-curated section of DFAM 3.7 which are not suited in my opinion to train a classifier. This section of DFAM is essentially composed of raw, unreviewed RepeatModeler runs. The consensus in these library can be complete (full-length element with exact boundaries), but in practice these libraries have redundancy, are fragmented, contain chimerical elements and contamination from gene families and non TE repetitive elements. All of these are classified with Repeatclassifier, an homology-based module that relies on the local DFAM installation present on the machine where the run is executed. The version and scope of the local DFAM library can vary widely (DFAM libraries are divided in partitions that can be individually installed). This means that the quality of the label attributed to the non-curated DFAM consensus is expected to be highly variable and in my opinion no suited to train a TE classifier.

Nevertheless, this should not prevent the authors to explore the current limitation of using a transformer based framework for TE classification. Whatever the results are without the un-curated DFAM consensus, the science behind it will be valid and will inform further work in the field. We can perhaps learn what classes or types of TEs are more likely to be recognized by such model architecture and identify areas were progress need to be made. Note that DFAM is now at 3.9 and more curated models have been added.

We are aware of these shortcomings. However, for some of TE types, the number of models in the curated Dfam database is really too small for the model training. This is obviously changing with every new release of Dfam. Hopefully, at some point, we will be able to train our models only on the curated dataset, which should significantly improve TEclass performance. In the meantime, we have to rely on an imperfect dataset.

There are also interesting points in this manuscript that need to be highlighted. I think that the evaluation experiment has some merit by recreating realistic use cases of building de novo libraries with Repeatmodeler and attempting classification. However, the procedure should be described more precisely: how does the HMMER parameters selected by the authors translate in terms of confidence in the classification? For instance, I would feel more confident with controls on how much a consensus is covered by a hit with a certain percent of identity to a curated DFAM element. Perhaps retaining best reciprocal matches between the Repeatmodeler2 library and DFAM would create a high quality set to evaluate?

We used an E-value cutoff of 1e-10 with nhmmscan. This is a very stringent threshold that keeps only strong homology matches and helps ensure high-confidence classifications. This step was mainly to verify the RepeatModeler2 classifications. Since we’re working with model organisms whose TEs are already in the Dfam database, this step was really just an extra precaution for the reference set we’re using to compare different models. This is not a reclassification step; it' s just a way to confirm the RM2 classification and remove any possible ambiguities, such as weak or conflicting hits.

Recently, a CNN-RNN method was published but failed to compare their results to TEclass2 (even though it was mentioned in introduction); including CREATE (https://academic.oup.com/bib/article/26/6/bbaf608/8324532) to the evaluation would be very informative. Overall it would be great to have a figure summarizing the comparisons between ML methods.

Thank you for the suggestion. Although we didn’t create such a figure, we summarized the ML comparison in the new Table 5. We hope this makes the presentation more clear now.

Finally, I think that the most important outcome of this work would be advising potential users on how to make the best use of their model in further research, whether in ways to improve the model or ways to use the model to improve classification of TE. I would like to stress that in my view it doesn't matter how good the model perform. Even negative results are informative for the TE community.

Thank you for these interesting suggestions. We have rewritten the “Conclusion” section to reflect these suggestions.

Round 2

Reviewer 1 Report

Comments and Suggestions for Authors

The authors have addressed all of my points. I just have a few minor comments:

First, I understand that it is currently not possible to address the issue of evaluating the use of the non-curated Dfam database. Thus, it would be helpful to indicate in the manuscript how many sequences are curated and how many are not. The potential species representation bias should also be discussed. All of this could be indicated as next improvement steps.

Second, the new Table 5 should be referenced in the text.

Author Response

The authors have addressed all of my points. I just have a few minor comments:

First, I understand that it is currently not possible to address the issue of evaluating the use of the non-curated Dfam database. Thus, it would be helpful to indicate in the manuscript how many sequences are curated and how many are not. The potential species representation bias should also be discussed. All of this could be indicated as next improvement steps.

Thank you for understanding. We added discussion on Dfam weaknesses (lines 447-457).

Second, the new Table 5 should be referenced in the text.

Thanks for pointing out this omission . It has been fixed, see line 409.

Reviewer 2 Report

Comments and Suggestions for Authors

I appreciate the response of the authors to my comments. I understand that at this time, the authors do not have the personnel available to carry-out extensive additional experiment. Nevertheless, I think it is worth publishing their finding upon considering the following:

- please strengthen the "Known limitations" section based on both reviewer's critiques, in particular the use of the non curated section of DFAM, the use of an older version, and the fact that not all species and type of TEs are represented homogeneously
- please include significant critical discussion about the field of ML-based classification of TE. In particular, please discuss two recent publications/preprint and how the approach differ from TEclass2: https://github.com/yiqichen-2000/BERTE and https://pmc.ncbi.nlm.nih.gov/articles/PMC12619909/

Author Response

I appreciate the response of the authors to my comments. I understand that at this time, the authors do not have the personnel available to carry-out extensive additional experiment. Nevertheless, I think it is worth publishing their finding upon considering the following:

- please strengthen the "Known limitations" section based on both reviewer's critiques, in particular the use of the non curated section of DFAM, the use of an older version, and the fact that not all species and type of TEs are represented homogeneously
- please include significant critical discussion about the field of ML-based classification of TE. In particular, please discuss two recent publications/preprint and how the approach differ from TEclass2: https://github.com/yiqichen-2000/BERTE and https://pmc.ncbi.nlm.nih.gov/articles/PMC12619909/ 

Thank you for your understanding and suggestions. 

We added discussion on the Dfam limitation and current ML-based tools (lines 447-461 and Table 6).